# Immunological Aspects of Infertility—The Role of KIR Receptors and HLA-C Antigen

**DOI:** 10.3390/cells13010059

**Published:** 2023-12-27

**Authors:** Anna Wasilewska, Marcelina Grabowska, Dominika Moskalik-Kierat, Martyna Brzoza, Piotr Laudański, Marzena Garley

**Affiliations:** 1Laboratory of Immunogenetics, University Clinical Center, Medical University of Warsaw, 02-091 Warsaw, Poland; wasilewska.anna1989@gmail.com (A.W.);; 2Department of Obstetrics, Gynecology and Gynecological Oncology, Medical University of Warsaw, 02-091 Warsaw, Poland; 3Women’s Health Research Institute, Calisia University, 62-800 Kalisz, Poland; 4OVIklinika Infertility Center, 01-377 Warsaw, Poland; 5Department of Immunology, Medical University of Bialystok, 15-269 Białystok, Poland

**Keywords:** infertility, infertility immunology, miscarriage, implantation, KIR, HLA-C, uNK, pregnancy

## Abstract

The mechanisms of immune tolerance of a mother against an antigenically foreign fetus without a concomitant loss of defense capabilities against pathogens are the factors underlying the success of a pregnancy. A significant role in human defense is played by killer immunoglobulin-like receptor (KIR) receptors, which regulate the function of the natural killer (NK) cells capable of destroying antigenically foreign cells, virus-infected cells, or tumor-lesioned cells. A special subpopulation of NK cells called uterine NK cells (uNK) is found in the uterus. Disruption of the tolerance process or overactivity of immune-competent cells can lead to immune infertility, a situation in which a woman’s immune system attacks her own reproductive cells, making it impossible to conceive or maintain a pregnancy. Since the prominent role of the inflammatory response in infertility, including KIR receptors and NK cells, has been postulated, the process of antigen presentation involving major histocompatibility complex (MHC) molecules (HLA) appears to be crucial for a successful pregnancy. Proper interactions between KIR receptors on female uNK cells and HLA class I molecules, with a predominant role for HLA-C, found on the surface of germ cells, are strategically important during embryo implantation. In addition, maintaining a functional balance between activating and inhibitory KIR receptors is essential for proper placenta formation and embryo implantation in the uterus. A disruption of this balance can lead to complications during pregnancy. The discovery of links between KIR and HLA-C has provided valuable information about the complexity of maternal–fetal immune interactions that determine the success of a pregnancy. The great diversity of maternal KIR and fetal HLA-C ligands is associated with the occurrence of KIR/HLA-C combinations that are more or less favorable for reproductive success.

## 1. Infertility

Infertility is defined by the World Health Organization (WHO) as the inability to achieve pregnancy for a period of 12 months, despite regular sexual intercourse (2–4 times a week), without using any contraceptive methods [1]. Infertility is estimated to affect 10–15% of couples of reproductive age worldwide (approximately 60–80 million people a year). In Poland, the problem of infertility affects approximately 1–1.5 million couples [2].

A distinction is made between primary and secondary infertility. Primary infertility affects women who have never had a clinically established pregnancy. Secondary infertility occurs in women who have had a clinically established pregnancy in the past but are unable to have another one [3]. Typically, an assessment for infertility is started when women under the age of 35 have been engaging in regular unprotected intercourse for one year. For women aged 35 and above, this evaluation commences after six months of such intercourse. Nevertheless, women aged 40 and older, as well as those with irregular menstrual cycles or identifiable infertility risk factors like endometriosis, a past history of pelvic inflammatory disease, previous pelvic surgeries, reproductive tract abnormalities, or known male fertility issues, may initiate the evaluation earlier [4].

Infertility is a medical, psychological, and social problem. It is also an important factor affecting a country’s demographics and macroeconomic indicators; therefore, it also indirectly affects the well-being of a society [5,6].

The diagnosis and treatment of infertility require a holistic and comprehensive approach to the problem, as it can have various causes: infectious, genetic, anatomical, hormonal, and immunological [7]. First of all, the diagnosis should include an evaluation of ovarian function (occurrence of ovulatory cycles), anatomy of the female reproductive system (patency of the fallopian tubes, evaluation of the uterine cavity), and semen examination (semen parameters). Diagnosis of the causes of infertility should be carried out simultaneously in both partners. It is reported that 35–40% of infertility cases in couples are caused by a female factor only, 20–40% by a male factor only, and in 20–30% of cases, the problems are on both sides [8]. The diagnosis of infertility of unexplained origin (unexplained, idiopathic) can be made when routine diagnostic tests do not reveal the cause of infertility. It must be expanded to include genetics, immunology, molecular diagnostics, or surgery. The diagnosis can affect up to approximately 20–30% of infertile couples and is due to the imperfections and diagnostic limitations of modern medicine. The potential causes of unexplained infertility include genetic as well as immunological abnormalities. Currently, there are three main therapeutic strategies: drug treatment, surgical management, or assisted reproductive technology (ART). ART techniques include methods related to in vitro handling of human gametes or embryos to achieve pregnancy. One of the most effective infertility treatments is in vitro fertilization. Couples usually opt for in vitro fertilization as a last resort, after trying all other possible treatments [4]. Despite significant advances in the diagnosis and treatment of infertility, including undergoing assisted reproduction procedures, many women experience recurrent implantation failure (RIF) after in vitro fertilization [9].

Infertility can have various causes, and immune problems represent one of the many factors that can contribute to difficulties in conceiving. Immune-related factors of infertility are focused on autoimmune disorders, where the immune system mistakenly targets the body’s tissues and may affect fertility. The next problem is related to physiological reaction–alloimmunity, which involves an immune response against non-self antigens, such as those of a partner during pregnancy. Incompatibility between the partners’ immune systems can lead to implantation failure or recurrent miscarriages. The mother’s active immune system may produce anti-seminal/sperm antibodies. Moreover, chronic inflammation of a woman’s reproductive organs can damage the fallopian tubes, ovaries, cervix, and uterus. Immunological factors at the implantation site influence the success of embryo implantation. Immune cells and cytokines are critical in the intricate balance required for a successful pregnancy [10,11,12,13].

This review discusses the immunogenetic causes of infertility related to KIR (killer immunoglobulin-like receptor) and HLA-C antigens (human leukocyte antigen), whose interactions may affect the course of a pregnancy. Recently, the important role of the inflammatory response in infertility, involving natural killer cells (NK cells) and certain T-cell populations, as well as the potential involvement of the KIR receptor and its ligands, have been pointed out.

## 2. Implantation

Proper implantation of the embryo affects the development and maintenance of the pregnancy. The physiological and molecular processes initiated during implantation for a successful pregnancy are complex but highly organized. These processes depend on the tolerance of the mother’s immune system to the fetus possessing foreign antigens of paternal origin without losing the mother’s ability to fight infection [14]. Immune factors, mainly NK cells, which are present in the uterus, have become increasingly important in the proper implantation of the embryo [15,16]. Pregnancy is an immune paradox and the only physiological situation in which allorecognition by NK cells occurs. The main tissue location where maternal allorecognition of the fetus occurs is in the uterus at the location of the placenta, where fetal trophoblast cells attack and mix with maternal leukocytes. At this site, maternal immune cells naturally contact the semiallogeneic placenta [17].

On the other hand, during the peri-implantation period, mechanisms related to arterial effects play an important role [14]. This stage is characterized by deep infiltration of the placenta by transforming the arteries and decidualization of the endometrium (changes occurring in the functional layers of the endometrium under the influence of the implanting blastocyst in response to the secretory activity of the corpus luteum). During this process, the endometrium transforms into the decidua. During implantation, the number of immune system cells, mainly NK cells, macrophages, dendritic cells, and T lymphocytes, in the decidua increases. Immune system cells initiate the formation and remodeling of blood vessels in the uterus and are responsible for maintaining immune balance. Their role is also to regulate the proliferation and differentiation of endometrial mesenchymal cells, which facilitates transformation into the decidua. The decidua cells regulate trophoblast invasion and initiate the process of angiogenesis [18,19,20].

Decidualization is always associated with the presence of a unique population of lymphocytes devoid of cytolytic activity, i.e., uterine natural killer cells (uNK), which represent a leukocyte population of 70% in the first trimester of pregnancy. Since uNKs are in direct contact with invasive fetal placental cells, they are involved in placentation and thus in fetal growth and development. The uNK cells are classified as endometrial NK (eNK) or decidua NK (dNK) cells. The uNK cells cluster in large numbers around trophoblast cells. The most likely function of uNK appears to be the regulation of blood flow to the intervillous space and the regulation of arterial transformation by the trophoblast. A physical barrier is formed between the mother and fetus, allowing normal fetal growth and development without exposing the mother to danger [21,22,23,24]. Reduced trophoblast invasion and vascular conversion in the decidua are believed to be the main causes of common pregnancy disorders. Implantation defects are reflected in aberrations, abnormal decidualization, placentation, and intrauterine growth of the embryo, manifesting as pre-eclampsia, miscarriages, and/or preterm births. It is believed that 75% of abnormally terminated pregnancies resulted from implantation failure [14].

Implantation of the embryo is a process that is hormonally controlled by ovarian-derived estrogen and progesterone, but locally produced signaling molecules such as cytokines, growth factors, homeobox transcription factors (transcription factors of HOX genes key in hematopoietic cell differentiation and proliferation), and lipid mediators also play a role [25]. Progesterone is responsible for preparing the endometrium for implantation and modifying the maternal immune response to prevent allogeneic fetal rejection. Progesterone inhibits the Th1-type cellular response by stimulating the synthesis of progesterone-induced blocking factor (PIBF) in circulating blood and at the level of the trophoblast, as well as by blocking the activity and proliferation of T lymphocytes and NK cells [26].

## 3. NK, uNK, and ILC Cells

NK cells are a population of lymphocytes with potent cytotoxic effects formed in the bone marrow. These cells also have immunoregulatory functions, including secreting gamma interferon (IFN-γ), granulocyte–macrophage colony-stimulating factor (GM-CSF), interleukins (IL-5, IL-10, IL-13), and chemokines, contributing to the formation of the innate nonspecific immune response [27,28]. NK cells account for approximately 10% of all lymphocytes circulating in the blood. NK subtypes are distinguished by the presence of characteristic surface markers. NK cells recognize structures on the surface of other cells called major histocompatibility complex (MHC) molecules, which come in two classes, I and II. Human MHCs have been named HLAs (human leukocyte antigens). Class I and II antigens are divided into several subclasses based on their structure and level of expression. Subclass Ia includes HLA-A, -B, and -C. These are the so-called “classical transplantation antigens” found on all nucleated cells of the human body, although their expression on the surface of some cells may be limited. Subclass Ib contains the nonclassical HLA-E, -F, and -G antigens. HLA-E antigens are expressed in most tissues, although their levels are lower than those of classical MHC. Subclass Ic has two antigens, MICA and MICB (MHC class I-related chains). HLA class I molecules, both classical and nonclassical, are recognized by NK cells. MHC class II molecules are also divided into two subclasses: “classical” (class II a)—HLA-DR, HLA-DP, HLA-DQ, and “nonclassical” (class II b)—DM and DO. MHC class II molecules are found only on specialized cells of the immune system; so-called antigen-presenting cells [29,30].

Cytotoxicity and cytokine production are regulated by a number of receptors that are present on the surface of NK cells. These receptors can be either activating or inhibitory, which consequently affects the function of these cells as well as the differentiation of normal own cells from altered/foreign cells. NK cells attack and kill cells that lack or poorly express the MHC molecule on the surface of virus-infected, cancerous, and allogeneic foreign cells. This happens according to the “missing self” hypothesis when there is no inhibitory signal. NK cells then engage only activating receptors, leading to the secretion of proinflammatory cytokines and consequent actions resulting in lysis of the target cell. Sparing of healthy body cells occurs due to the presence of inhibitory receptors. Since each NK cell has its own repertoire of inhibitory and activating receptors, the final cytotoxic effect depends on the resultant activating and inhibitory signals after binding to the appropriate ligands [31,32,33,34,35].

The results of recent studies prove the existence of subpopulations of dNK cells. These populations differ not only in their numbers and surface marker profiles, but also in their biological roles. Figure 1 summarizes the most important information currently available regarding dNK cells [36].

NK cells in normal pregnancy are not cytotoxic, and their function is to produce cytokines and growth factors that promote pregnancy development. It is the activation, rather than the inhibition, of uNK cell activity that is essential for normal pregnancy, due to the secretion of substances necessary for placental development [37,38]. Figure 2 shows a schematic presentation of HLA antigens with the participation of KIR receptors in normal and abnormal pregnancies.

There is an innate lymphoid cell (ILC) group that plays a crucial role in the early defense against pathogens and in maintaining tissue homeostasis. ILCs are part of the innate immune system, which provides non-specific responses to pathogens without prior exposure. ILCs lack specific antigen receptors (TCR or BCR), meaning they do not recognize specific antigens, but they do respond to signals from the microenvironment. There are three main subsets of ILCs with distinct functions. The ILC1 type is involved in responses against intracellular pathogens, particularly viruses, by producing cytokines that support immune responses. T-bet+/Eomes+ NK cells produce perforin, granzyme, TNF-α, and IFN-γ. T-bet + ILC1 cells produce TNF-α and IFN-γ. ILC2 type (GATA3 + ILC2) is associated with allergic reactions and defense against parasitic infections. Their cytokines, IL-4, IL-5, IL-9, IL-13, and Areg, support inflammation and activate other immune cells. The ILC3 group plays a part in defense against bacterial and parasitic infections, particularly in the gut. They regulate mucosal barrier function and help maintain homeostasis. RORyt + ILC3 produce IL-17, IL-22, and GM-CSF. RORyt + LTi produce IL-17, TNF-α, IFN-γ, GM-CSF, RANK, and lymphotoxin [39].

ILC1 and uNK are distinct types of immune cells, with some similarities and differences. They are both part of the innate immune system and produce IFN-γ and TNF-α. The uNK cells play a crucial role in the regulation of immune responses in the uterus and tissue remodeling during pregnancy. They contribute to the maternal immune tolerance to the fetus. On the other hand, ILC1 cells are located in various tissues, including mucosal surfaces, and are not specifically associated with the uterus. ILC1s are generally involved in the immune response against intracellular pathogens, especially viruses, with a broader role beyond the reproductive system.

The exact contribution of uNK cells and whether their absence would lead to abnormal pregnancies in humans or animals is not fully understood. Some research has suggested that altering the number or function of uNK cells can affect fetal development and placental formation in mice [40]. In humans, the role of uNK in pregnancy is challenging to study directly due to the complexity of the human reproductive system and ethical considerations. In observations of humans, two female JAK3-deficient patients had no signs of fertility disorders and had healthy babies with normal birth weight and no other pathological symptoms following clinical event-free pregnancies [41]. The immune system during pregnancy involves interactions between various immune cell types. The potential absence of uNK cells might be compensated for by other immune cells or mechanisms.

## 4. Mechanisms of Immunological Tolerance

The mechanisms of immune tolerance and control of the balance between the maternal and fetal organism involve fetal MHC antigens, the cellular and functional barrier of the placenta, and suppression of the maternal immune response. Disruption of any of the above-mentioned immune processes can impede the development of pregnancy [42].

### 4.1. Expression of Fetal MHC Antigens

A successful pregnancy requires a proper relationship between maternal and fetal cells. Tolerance mechanisms in the mother enhance her tolerance to fetal cells but must ensure proper protection against potential infection and pregnancy (fetus) loss. In physiological pregnancy, there is cell migration in both directions (mother–fetus). Migrating cells can induce maternal and fetal T cell cytotoxicity, directed against both paternally and maternally inherited alloantigens. These include specific HLA leukocyte antigens (MHC) and so-called minor histocompatibility alloantigens (mHAG) [43]. MHC molecules are responsible for the presentation of foreign antigens to immune cells. The mHAG antigens are derived from functional proteins and can trigger an immune response due to allelic differences between individuals, usually single-nucleotide polymorphisms, insertions, deletions, or the presence of an antigen on the Y chromosome. Released into the circulation, fetal mHAG can be phagocytosed by maternal dendritic cells and presented to CD8+ T cells, triggering an immune response to antigenically foreign fetal [44].

Regulatory T cells and uNK cells are mainly involved in the development of tolerance of the mother’s immune system to the highly immunogenic paternal antigens present in fetal tissues. These cells take an active part in embryo implantation and placenta formation. After fertilization, the blastocyst activates the formation of the chimeric placenta, which is formed from the mother’s decidua cells and the fetal trophoblast, implanting in the uterine wall. Trophoblast invasion is an inflammatory response that is under the control of uNK cells. Unlike NK cells in the peripheral blood, which effectively kill target cells, uterine uNK cells usually do not show cytotoxicity to the fetus or placenta. They differ phenotypically from classical NK cells and show expression of KIR and FcγR (CD16) receptors (the receptors for the Fc of immunoglobulins) [45,46,47].

One of the mechanisms regulating placenta formation and implantation of the fertilized egg is the expression of the classical HLA class I antigen (HLA-C) by cells of the invasive trophoblast. This antigen is essential for the promotion of angiogenesis involving uNK cells. In contrast, the trophoblast lacks expression of other classical MHC class I and II antigens, which enables it to maintain tolerance to fetal paternal antigens. The inhibition of natural cytotoxicity of uNK cells is provided by the expression of nonclassical HLA-E and HLA-G molecules. Immunoglobulin-like receptors KIR are involved in the recognition of HLA-C and HLA-G molecules. Depending on their structure and interaction with ligands, KIR receptors can either activate or inhibit the NK cells in utero affecting the production of cytokines, chemokines, and growth factors necessary for normal embryo implantation and placental development. Inappropriate uNK activation, whether excessive or insufficient, can lead to pregnancy disorders such as recurrent miscarriages, premature births, and pre-eclampsia [45,48,49].

### 4.2. Maternal–Fetal Circulation Barrier

The maternal–fetal circulation barrier provides the basis for immune tolerance and protection of the fetus. In a physiological pregnancy, there is a lack of response to fetal antigens with a normal response to pathogen antigens—a mechanism of immune facilitation (protective effect of anti-HLA antibodies for the survival of antigenically distinct cells). The trophoblast produces exosomal vesicles with paternal HLA antigens, which reach the maternal circulation and local lymph nodes, where they are absorbed by maternal macrophages [50]. There, antigen presentation in the context of the MHC class II system and recognition of paternally derived HLA antigen determinants by Th2 lymphocytes takes place, B lymphocytes are activated, and anti-HLA antibodies are produced, which then return via the blood route to the endometrium. This process occurs in the early stages of pregnancy when the blastocyst implants in the uterine wall. Anti-HLA antibodies can bind antigens and form insoluble immune complexes. The intensity of this reaction depends on the concentration gradient of antigens and antibodies. Most efficiently, complexes are formed in the equilibrium zone (halfway between the implanting blastocyst and the lymph node), which is a mechanism for protecting the embryo from antibody attacks. As a result of trophoblast cell proliferation, the concentration of soluble paternal HLA antigens secreted outside the uterus is reduced, and the zone of immune complex formation shifts toward the placenta. When the zone of the quantitative balance of fetal antigens and maternal antibodies is within the trophoblast, the precipitates of the formed immune complexes seal the placenta. This leads to the formation of a maternal–fetal circulation barrier [51,52,53].

### 4.3. Suppression of Maternal Immune Responses

Maternal CD4+CD25+ regulatory T cells (Treg) play an important role in creating a state of tolerance to fetal antigens, especially in the first weeks, reaching a level of 15% among decidua lymphocytes in physiological pregnancy. Treg CD4+CD25+ deficiency is associated with failure of blastocyst implantation in the early stages of pregnancy. In addition, trophoblast cells also express FasL (Fas ligand), which induces apoptosis of activated CD8+ cytotoxic T cells possessing the Fas receptor through a clonal deletion mechanism, resulting in the extinction of the unwanted immune response directed against fetal antigens [54,55,56].

Suppression of the maternal immune response is also associated with anti-inflammatory cytokines (e.g., IL-10) triggering “Th2-dependent” humoral immunity. However, certain proinflammatory cytokines are essential for successful implantation [57]. During trophoblast invasion and during delivery, the Th1 response and proinflammatory cytokines (TNF-α, and interleukins: IL-1, IL-2, IL-12, IL-15, IL-18) predominate. In contrast, in the second and third trimesters of pregnancy, anti-inflammatory cytokines (including IL-4, IL-5, IL-6, IL-10, IL-13, and GM-CSF factor) predominate, mainly to facilitate fetal growth and maintenance of the pregnancy. At the end of the third trimester of pregnancy, there is a renewed predominance of Th1-type responses, as the infiltration of immune cells into the uterine myometrium is important in this phase to promote uterine contractions, labor, and placental separation [58,59]. The differentiation of Th lymphocytes is influenced by hormones. Low estradiol levels during pregnancy promote Th1 responses, while high levels promote Th2 responses. In contrast, elevated levels of progesterone induce the secretion of Th2 cytokines and inhibit the secretion of Th1 cytokines. Pregnancy failure may be associated with Th1-mediated cytokines triggering cellular immunity. The predominance of Th1 over Th2 responses may be associated with various complications of pregnancy, including recurrent miscarriages, pre-eclampsia, and embryo implantation failure after in vitro fertilization [60].

## 5. Structure of KIR Receptors

NK cell receptors, which detect MHC class I molecules on the surface of target cells, belong to two families of proteins: lectin-like receptors and the immunoglobulin-like receptor superfamily. KIR receptors play an important role in the control of NK cell function. They are receptors present on the surface of natural cytotoxic cells and certain subpopulations of T cells: γδ, αβ, CD8+, and CD4+ [61,62,63]. Because both KIR receptors and MHC class I antigens are highly polymorphic, their interactions differentiate and individualize the NK cell response [64].

The nomenclature of KIR receptors is created based on their structure, and the number of extracellular protein domains is similar to the immunoglobulins: 2D denotes two domains (D1 and D2), 3D denotes three domains (D0, D1, D2), and the length of the cytoplasmic tail is L for a long tail and S for a short tail [65]. In addition, in humans, there are several types of each of these molecules, each encoded by separate genes. In the notation, the last digit in the name of each receptor indicates the number of the gene encoding the protein in that structure [66,67].

KIR receptors are characterized by the presence of extracellular domains, a trans-membrane domain, and a cytoplasmic tail. Due to the structure of the cytoplasmic part, KIR receptors can be activating or inhibitory. Typically, inhibitory KIRs have a long cytoplasmic tail (KIR2DL), while activating ones have a short cytoplasmic tail (KIR2DS) [65].

The genes encoding KIR receptors are located on chromosome 19 in the q13.4 region within the leukocyte receptor complex (LRC). A total of 17 KIR genes were identified: 6 activating (*KIR2DS1*, *KIR2DS2*, *KIR2DS3*, *KIR2DS4*, *KIR2DS5*, *KIR3DS1*), 8 inhibitory (*KIR2DL1*, *KIR2DL2*, *KIR2DL3*, *KIR2DL5A*, *KIR2DL5B*, *KIR3DL1*, *KIR3DL2*, *KIR3DL3*), 2 pseudogenes (*KIR2DP1* and *KIR3DP1*), and *KIR2DL4*, which has both activating and inhibitory potential [68,69]. According to the Immuno Polymorphism Database, 1632 KIR alleles have been identified to date (https://www.ebi.ac.uk/ipd/imgt/hla/, (data as of 17 May 2023). KIR genes show high allelic (large number of alleles) and haplotype polymorphisms (different number of genes in different individuals). They are present in all human populations, but with different relative frequencies, suggesting that they have different functional properties. Because people differ in both the number and types of KIR receptors, susceptibility to KIR-related diseases is different among different individuals [9,64,67,70].

There are two haplotypes: A and B. Both haplotypes consist of four genes, the so-called “framework genes”: *KID3DL3*, *KIR3DL2*, *KIR3DP1*, and *KIR2DL4*, which are found in almost all individuals. Haplotypes A have fixed gene contents. The following are genes encoding only inhibitory receptors: *KIR2DL1*, *KIR2DL3*, *KIR3DL1*, *KIR2DP1*. Only one gene encodes the activating receptor, *KIR2DS4*. Haplotypes B, compared to haplotypes A, are much more diverse in terms of the total number of genes, their combinations, and the number of activating KIR genes; therefore, they can show activating activity after interacting with the corresponding ligand. Haplotypes B are characterized by the presence of at least one of the following genes: *KIR2DS1*, *KIR2DS2*, *KIR2DL2*, *KIR2DL5A*, *KIR2DL5B*, *KIR2DS3*, *KIR2DS5*, or *KIR3DS1*. In Poland, the majority of the population has haplotype B (69.9%), compared to 29.7% with haplotype A (http://www.allelefrequencies.net/ (data as of 17 May 2023)) [71,72].

Since KIR genes have two regions, namely centromeric (Cen) and telomeric (Tel), they are also divided according to the presence of A and B haplotypes in the centromeric or telomeric region: CenA has *KIR2DL3*; CenB has *KIR2DL2* and *KIR2DS2*; TelA has *KIR3DL1* and *KIR2DS4*; and TelB has *KIR2DS1* and *KIR3DS1* [73]. The beginning and end of the regions are determined by the “framework” genes—*KIR3DL3* and *KIR3DP1* define the centromeric region, while *KIR2DL4* and *KIR3DL2* define the telomeric region. Haplotype A includes the CenA and TelA regions, while haplotypes B includes various combinations, such as CenA/TelB, CenB/TelA, or CenB/TelB [74].

A person who has both copies of haplotype A is considered KIR AA, and a person who has haplotype B genes is considered KIR Bx (AB, BB). In other words, a person’s KIR genotype can be A/A (0–1 activating KIR), A/B (1–6 activating KIR), or B/B (3–10 activating KIR). The AA genotype is the set of genes for the greatest number of inhibitory receptors with no activating genes. The intensity of inhibition decreases as the number and expression of KIR receptor-activating genes increases [9]. It should be emphasized that NK cell function depends on the balance between inhibitory and activating receptors [75,76]. The ligands for inhibitory and activating KIR receptors are molecules of the major human class I tissue compatibility system: HLA-A, HLA-B, and HLA-C allotypes. It has been shown that activating KIR receptors can bind HLA class I, but more weakly than inhibitory ones, meaning that inhibition predominates over activation [71].

## 6. HLA-C on Trophoblast Cells

Trophoblast cells have a unique, tissue-specific HLA class I expression pattern that includes invariant HLA-G and HLA-E and polymorphic HLA-C but not highly polymorphic HLA-A and HLA-B. Therefore, of the HLA class I molecules expressed on the trophoblast, only HLA-C shows the variability necessary for fetal alloantigen [77].

HLA-C molecules are polymorphic heterodimers, with 7761 alleles known to encode 4311 proteins (IPD-HLA database, 25 May 2023). HLA-C may specialize in binding and presenting specific peptides, likely derived from viral proteins. They are also the most important ligands of KIR receptors, which can activate (*KIR2DS*) or inhibit (*KIR2DL*) the innate response [72].

The HLA-C gene exists in two allotypes, C1 and C2, depending on the presence of asparagine or lysine at position 80 of the α domain of HLA-C [78]. In other words, the two mutually exclusive HLA-C epitopes are defined by a different amino acid residue and are recognized by different KIRs. The inhibitory *KIR2DL1* and activating *KIR2DS1* recognize the C2 (lysine 80) epitope, the inhibitory *KIR2DL3* recognizes the C1 (aspartate 80) epitope, and the inhibitory *KIR2DL2* essentially recognizes C1, with some cross-reactivity with C2 [64]. The HLA-C C1 group includes the following: HLA-Cw1, Cw3, Cw7, Cw8, Cw12, Cw14, and Cw16. The HLA-C C2 allele group includes the following: HLA-Cw2, Cw4, Cw5, Cw6, Cw7, Cw12, Cw15, Cw16, Cw17, and Cw18 [73,79]. The difference between the two KIR haplotypes with respect to HLA-C is the presence of the *KIR2DS1* gene in haplotype B [80].

Comparing the A and B KIR haplotypes, it appears that there are KIR-encoding genes in each Cen and Tel region that can bind different HLA-C allotypes with varying affinity. This means that in each individual, the strength of the systemic NK response will depend on the presence of the *KIR2DL1/2/3* inhibitory allele and the *KIR2DS1/S4* activating allele, as well as on the presence of the C1/C2 alleles that the individual inherits [21].

The centromeric region of haplotype A, characterized by *KIR2DL1* and *KIR2DL3*, binds HLA-C2 and C1, respectively. Centromeric region B, defined by *KIR2DS2* and *KIR2DL2*, binds C1 and some C2 allotypes. Telomeric region A (*KIR3DL1* and *KIR2DS4* are located there) binds some C1 and C2 allotypes, and telomeric region B (*KIR3DS1* and *KIR2DS1*) binds all C2 allotypes. At the same time, the interaction of *KIR2DL1* with C2 is stronger than with *KIR2DL2* or *KIR2DL3* with C1. The HLA-C2 group of molecules is more strongly activated by effector response inhibitory genes, and inhibitory KIRs show greater variability than activating ones. The KIR genes in each of these regions are in strong coupling disequilibrium; that is, the immune system’s balanced response to the presence and action of stimulants and the balance between immune response and tolerance [81].

HLA-C exhibits significant genetic diversity among different populations, and variations can influence not only the immune response but also reproductive success [22]. There are some general points regarding racial and ethnic differences in HLA-C. Different ethnic groups can have distinct allele frequencies and haplotypes, e.g., the HLA-C alleles in individuals of European descent may differ from those of African or Asian descent.

## 7. KIR2DL4 and HLA-G

The *KIR2DL4* receptor is an unusual receptor in terms of structure, expression, localization, and function [82]. It has mixed properties, activating and inhibitory, and binds a small polymorphic so-called nonclassical molecule, HLA-G, found only on trophoblast cells. The combination of the KIR2DL4 receptor and HLA-G triggers immune tolerance [83]. HLA-G antigens have a limited ability to present antigens, thereby inhibiting its recognition as foreign, preventing the cytotoxic action of NK cells during placental vascular adaptation, and playing a positive role in the final success of the pregnancy [7]. The interaction of the HLA-G antigen serves to inform the maternal body of the presence of the fetus rather than its foreignness. In contrast, the polymorphic HLA-C antigen inherited from the child’s father, also present on trophoblast cells, stimulates the NK cells present in the uterus with its “foreignness” to secrete the cytokines important for implantation and proper nutrition of the fetus [84,85].

## 8. KIR Can Play an Inhibitory and Activating Role

Inhibitory KIRs have an amino acid motif called immunoreceptor tyrosine-based inhibitory motif (ITIM) which inhibits NK function. Inhibitory receptors have a long cytoplasmic tail (L) and one or two ITIM inhibitory motifs. Activating KIRs contain positively charged lysine residues in the transmembrane region and have shortened cytoplasmic tails devoid of ITIM. They do not have any amino acid motifs associated with cell activation, but they form complexes with so-called DAP12 adapter proteins, which have an immunoreceptor tyrosine-based activatory motif (ITAM) amino acid motif in the cytoplasmic region. After ligand binding by KIR, the DAP12 protein is recruited via ITAM, which initiates intracellular signaling pathways and cytokine secretion and induces NK cell activation [66,71,86,87].

## 9. Research Review

As mentioned above, HLA class I molecules are ligands for the KIR receptors found on NK cells. HLA–KIR ligation either stimulates or inhibits the ability of NK cells to cytolyze foreign cells, including tumor cells and allogeneic cells, depending on the type of KIR receptor stimulated and the affinity of the ligation [88]. The discoveries of KIR and HLA-C combinations underscore the complexity and importance of specific combinations in maternal–fetal immune interactions. The great diversity of maternal KIR and fetal HLA-C ligands means that certain KIR/HLA-C combinations may be more or less beneficial for reproductive success. Both KIR and HLA-C are highly polymorphic; therefore, each pregnancy will be characterized by specific combinations of maternal KIR and fetal HLA-C variants. These interactions, as well as the properties of HLA-C molecules, are crucial to understanding how the maternal immune system can influence reproductive success [77].

A Polish study in a group of 1064 men examined which potential KIR and HLA-C genes/receptors and their combinations may affect infertility, and whether KIR and HLA-C genes affect sperm parameters. The KIR profile and frequencies of HLA-C genotypes were compared in men with various sperm abnormalities undergoing in vitro fertilization (IVF). Significant differences in the KIR gene profile were found between men who had children from natural conception and men who participated with their partners in in vitro fertilization. More of the men reporting to the infertility clinic had the activating receptor genes *KIR2DS2*, *KIR2DS3*, as well as the CenAB + BB genotype. The AA genotype, as well as a combination of the AA genotype and HLA-C2 allotype, was likely to be more common in the fertile group. The worst combination of KIR and HLA-C for male infertility is CenAB/TelBB/C2+, where the *KIR2DS1*, *KIR2DS5,* and *KIR3DS1* genes are also present. It has been proven that KIR–HLA-C interactions can affect male infertility and indirectly couple infertility. Fertile men differ in their KIR gene profile and KIR–HLA-C combinations from men entering IVF treatment. Potential integrations between activating KIR receptors in CenAB + BB carriers and HLA-C allotypes may influence male infertility, while carriers of the KIR CenAA genotype do not experience problems with conception [61].

As shown in other Polish studies, the KIR AA genotype appears to be protective. Its presence was significantly higher in men from fertile controls than in men participating in in vitro fertilization. The observed effect in men was quite the opposite of what was observed in women. KIR genes from the AA telomeric region are predisposed to recurrent implantation failure after in vitro fertilization (RIF). Women with two *KIR A* haplotypes (*KIR AA* genotype) in combination with *HLA*-*C2* in the fetus have been shown to be at increased risk for placental abnormalities. The risk is greatest if HLA-C2 is inherited by the fetus from the father and the mother does not have HLA-C2 [45,89].

Similarly, a British study regarding maternal KIR and its interaction with HLA-C showed that women with the KIR AA genotype are at an increased risk of pregnancy disorders. Moreover, when confronted with additional alleles of the HLA-C2 group representing the paternal allogeneic C2 ligand on invasive fetal trophoblast cells, there is a higher risk of pregnancy complications. It has also been proven that the presence of the activating gene *KIR2DS1* in the telomeric region in haplotype B (when the fetus is HLA-C2), provides protection against pregnancy disorders in recurrent miscarriages, preeclampsia, or intrauterine fetal growth restriction. The KIR B haplotype is beneficial because there is a protective effect against complications when centromeric (Cen-B) and/or telomeric (Tel-B) genes are present within a region containing *KIR2DS1*, the activating receptor of the C2 groups. Studies have demonstrated how beneficial the KIR2DS1-HLA-C2 interaction is for placenta formation. It has also been shown that pregnancies with *Cen-B* KIR genes alone, without the Tel-B KIR region, have a lower risk compared to pregnancies with KIR AA [22,77,90].

Studies have shown an association between KIR, HLA-C genes, and predisposition to RIF. Carriers of genes for KIR inhibitory receptors from the AA telomeric region and HLA-C2C2 have been shown to have a higher risk of infertility and recurrent embryo implantation failure. Maternal possession of KIR genes from haplotype A is disadvantageous for embryo implantation. This is because once the ligand HLA-C2 is recognized on the trophoblast, KIR receptors will inhibit NK cells. The absence of genes for activating receptors means there is no activation of NK cells to secrete the cytokines and growth factors necessary for normal fetal development. The mother’s KIR AA genotype is a risk factor when the fetus has the HLA-C2 allele, as strong inhibition of NK cells can occur. This is because KIR2DL1 binds to the HLA-C2 epitope, and their interaction is stronger than the interaction of KIR2DL2 or KIR2DL3 with HLA-C1. In contrast, the reverse is true for women with the Bx (BB or AB) genotype. Haplotype B is characterized by the presence of the most genes for activating KIR. Even if the embryo inherits HLA-C2 from the father, the strong inhibition that results from the interaction between KIR2DL1 and HLA-C2 will be offset by activating signals from KIR-activating receptors, resulting in better trophoblast invasion. Instead of KIR2DL1, KIR2DS1 may bind to the HLA-C2 molecule. However, this inhibitory receptor has a higher affinity for the ligand than KIR2DS1. It all depends on which receptor is expressed on the cell surface. In addition, men with a set of genes for KIR-activating receptors (CenAB and CenBB) have a higher probability of infertility, which can affect a couple’s infertility. It was also proven that men entering IVF treatment had a higher prevalence of *KIR2DS2*, *KIR2DS3*, *KIR2DL2*, and *KIR2DL5* genes when RIF occurred, which shows a link to infertility [45,89].

Similar conclusions were reached by researchers in a 2015 paper on male factors in infertility. They observed that pre-eclampsia is more common in women with the KIR AA haplotype, especially with strong inhibition of uNK cell activity and HLA-C2 antigens in the fetus [7].

A study by Xiong et al. highlighted the beneficial relationship shown by the KIR2DS1–HLA-C2 interaction for placenta formation. Both inhibitory KIR2DL1 and activating KIR2DS1 bind HLA-C2, but they increase the risk or protection against pregnancy disorders, respectively. This suggests that in women with KIR AA, binding of inhibitory KIR2DL1 to HLA-C2 on the trophoblast results in impaired trophoblast invasion. In contrast, women with the KIR B haplotype have activation of the *KIR2DS1* gene, which also binds to HLA-C2. This can increase placental invasion and pregnancy development. Activation of KIR2DS1 on uNK by HLA-C triggers a response that enhances trophoblast migration and alters the response of uNK, which co-expresses the “risk” inhibitory receptor KIR2DL1. In general, clinical failures associated with placental abnormalities can manifest as pre-eclampsia, intrauterine growth restriction (IUGR), or recurrent miscarriages. In pregnancies with these clinical problems compared to controlled pregnancies, the frequency of the maternal KIR A/A genotype is increased in association with a fetus carrying the *HLA-C2* gene. The KIR B haplotype provides protection, particularly to those KIRs located in the telomeric B region (*KIR2DS1*, which encodes the activating KIR for C2 allotypes, is located there) [90]. There are also reports that the presence of the C2 allele in the fetus of a negative mother (with C1/C1) poses a risk of pregnancy failure that is higher than that of a C2/X mother [91].

A study by Akbari et al. demonstrated an association between recurrent miscarriages and the presence of KIR genes in women. The *KIR3DL1* gene was a protective factor, and the *KIR2DS2* and *KIR2DS3* genes appeared to be risk factors [83]. 

According to a study by Würfel et al., in 78% of patients with more than five RIFs, KIR receptor typing revealed the absence of three activating receptors (KIR2DS1, KIR2DS3, and KIR3DS5). The authors suggested that in this group of patients, in whom there is a defect in maternal–embryonic communication during implantation, the use of granulocyte colony-stimulating factor (G-CSF) is an extremely promising adjuvant treatment option [92].

Research linking KIR/HLA-C gene interactions to disease occurrence poses many difficulties. This is because important *KIR* genes (e.g., *KIR2DL1*) are present in most populations and are difficult to link to a disease entity. This high variability makes it difficult to carry out a straightforward analysis of the data, but KIR/HLA-C interactions have been demonstrated in several conditions. These include infectious diseases such as HCV (chronic hepatitis C virus infection), a history of HBV, vertical transmission of HIV-1, as well as autoimmune and inflammatory conditions such as acute kidney transplant rejection, ankylosing spondylitis, Crohn’s disease, ulcerative colitis, type 1 diabetes mellitus, idiopathic bronchial dilatation, multiple sclerosis, primary sclerosing cholangitis, psoriasis, psoriatic arthritis, rheumatoid arthritis and vasculitis, cancer (cervical cancer, malignant melanoma, chronic myelogenous leukemia), and diseases associated with reproductive disorders (pre-eclampsia, recurrent miscarriages, intrauterine hypotrophy) [93,94,95].

There are reports that KIR A haplotypes are beneficial in NK cell responses to infectious diseases, while the KIR B haplotype is associated with autoimmune diseases. Theoretically, it can be hypothesized that people with the KIR A/A genotype have more responsive NK cells, and their own mechanism of HLA antigen recognition is impaired in disease due to strong inhibitory signals received during NK development [21]. The fact that KIR A and B haplotypes have complementary functions in immunity and reproduction may explain why they are found in all human populations [96].

## 10. Summary

Infertility and difficulty getting pregnant are now growing clinical and social problems. The proper development of pregnancy depends on the tolerance of the mother’s immune system to the embryo. Currently, many studies are focusing on the role of NK cells in physiological and pathological pregnancy, as NK cells play a key role in fetal–maternal immune tolerance. Mechanisms in the placenta are regulated by interactions between maternal killer immunoglobulin-like receptors (KIRs) located on uNK cells and their ligands: HLA-C molecules presented by fetal trophoblast cells. Presentation of antigenic peptides to the KIR receptors present on fetal NK cells, which, after recognizing HLA class I molecules on trophoblasts, can stimulate or inhibit NK cells to secrete cytokines and growth factors and exhibit low-level cytotoxicity, affecting normal embryo implantation. The balance of all activating and inhibitory signals in NK cells in the fetus and trophoblast is crucial for a successful pregnancy. The role of uterine NK cells and KIR receptors and the search for their association with normal fetal development and failure are gaining increasing interest among researchers. The diversity of maternal KIR receptors and HLA-C allotypes on the trophoblast means that some combinations may be more favorable for embryo implantation and pregnancy maintenance due to the signal the NK cell receives. Research on polymorphic KIR genes and their HLA-C ligands is justified in the context of pregnancy and represents the future of infertility diagnoses.

## Figures and Tables

**Figure 1 cells-13-00059-f001:**
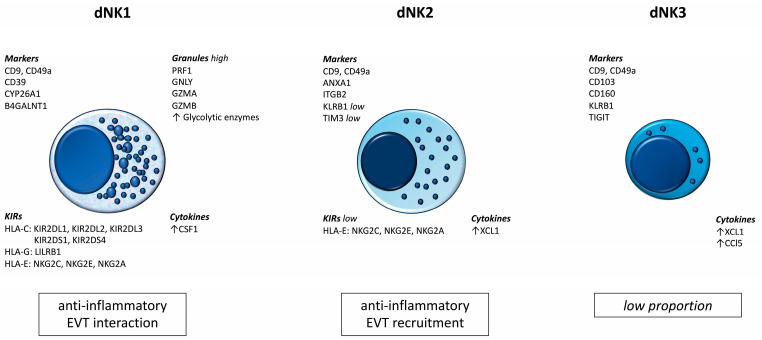
Subpopulations of dNK cells.

**Figure 2 cells-13-00059-f002:**
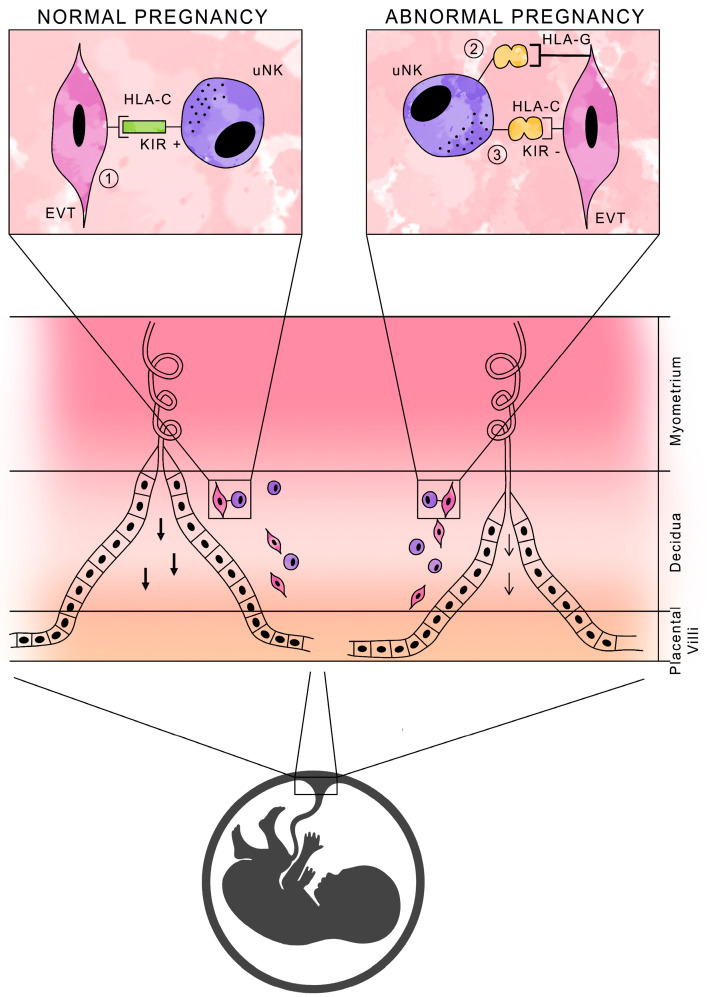
Schematic of HLA antigen presentation with KIR receptor participation during pregnancy.

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
