# Peer review of "Immunological Aspects of Infertility—The Role of KIR Receptors and HLA-C Antigen"

_cells, 2023, doi:10.3390/cells13010059_

Round 1
Reviewer 1 Report
Comments and Suggestions for Authors
The review by Wasilewska et al is well-written and in-depth discussion of how uterine NK cell genetics influence pregnancy. The authors propose that uNK cells promote or restrict fertility dependent upon genotype interactions. Specifically, which sex has which KIR and HLA-C haplotype(s) determines fertility aspects like trophoblast invasion and immune activity.
The review is excellently written, and as far as this reviewer can tell faithfully represents the literature. Points of suggestion are as follows:
Could the authors expand on important cell subsets: For example, innate lymphoid cells (ILCs) are not mentioned. Could the authors succinctly define the difference between a uNK and ILC1? Also, could the authors mention recent work showing three major NK cell subsets (e.g. PMID: 30429548)
The authors state that during pregnancy IFN-g comes from Th1 cells. Unless mistaken, this reviewer was led to believe that uNK are the major contributors of IFN-g? Our understanding is the mice without NK cells have poor vascular remodeling, which is rescued when exogenous IFN-g is delivered to the placenta. Unless mistaken, could the authors amend lines 248 and 135?
Is there strong evidence that humans (e.g. PMID: 27618553 ) and model animals (e.g. lots of mouse studies) without uNK cells can have normal pregnancy? If relevant, could the authors include?
Author Response
Respond to Reviewer Comments and Suggestions:
In the introduction, we would like to thank all reviewers for reviewing our manuscript and for their valuable comments and suggestions. We revised the manuscript according to the instructions. Changes are marked in green.
Reviewer 1.
The review by Wasilewska et al is well-written and in-depth discussion of how uterine NK cell genetics influence pregnancy. The authors propose that uNK cells promote or restrict fertility dependent upon genotype interactions. Specifically, which sex has which KIR and HLA-C haplotype(s) determines fertility aspects like trophoblast invasion and immune activity.
The review is excellently written, and as far as this reviewer can tell faithfully represents the literature.
Thank you very much for your positive opinion.
Points of suggestion are as follows:
Could the authors expand on important cell subsets: For example, innate lymphoid cells (ILCs) are not mentioned. Could the authors succinctly define the difference between a uNK and ILC1? Also, could the authors mention recent work showing three major NK cell subsets (e.g. PMID: 30429548)
Thank you for your tips.
A figure was prepared based on the referenced publication, summarizing dNK cell populations. We mentioned the ILC population of cells. And define the differences between uNK and ILC1.
The authors state that during pregnancy IFN-g comes from Th1 cells. Unless mistaken, this reviewer was led to believe that uNK are the major contributors of IFN-g? Our understanding is the mice without NK cells have poor vascular remodeling, which is rescued when exogenous IFN-g is delivered to the placenta. Unless mistaken, could the authors amend lines 248 and 135?
Changed as suggested.
Is there strong evidence that humans (e.g. PMID: 27618553 ) and model animals (e.g. lots of mouse studies) without uNK cells can have normal pregnancy? If relevant, could the authors include?
The results of other authors from the indicated publication were taken into account.
Thank you for your valuable time reviewing our manuscript.
Reviewer 2 Report
Comments and Suggestions for Authors
The authors described about KIR and HLA-C.
1. It would be even more useful to discuss racial differences in HLA-C.
2. Mistype? Lane 482, KIR3DS5→KIR2DS5?
Author Response
Respond to Reviewer Comments and Suggestions:
In the introduction, we would like to thank all reviewers for reviewing our manuscript and for their valuable comments and suggestions. We revised the manuscript according to the instructions. Changes are marked in green.
Reviewer 2.
The authors described about KIR and HLA-C.
- It would be even more useful to discuss racial differences in HLA-C.
The existence of racial differences in HLA-C was mentioned. These differences are not described in detail because they are not the main topic of the manuscript and may constitute the title of a new review paper.
- Mistype? Lane 482, KIR3DS5→KIR2DS5?
That's correct.
Thank you for your valuable time reviewing our manuscript.
Reviewer 3 Report
Comments and Suggestions for Authors
This review manuscript by Anna Wasilewska and colleagues described the function of KIR and HLA-C in infertility. The topic is of great interest and importance. The authors have reviewed a suite amount of diverse literature more or less related to the topic. Some sections of the manuscript are well-written. However, some may need to improve, such as:
1. Section 1 may need to highlight the reasons for infertility and specific reasons for immune problems.
2. Specifically, during pregnancy, most NK cells are called uNK; section 2 needs to describe the differences and functions of uNK cells.
3. Section 5 is better combined with sections 6, 7, 8, and 9 to focus on the function of KIR and HLA-C in infertility.
Author Response
Respond to Reviewer Comments and Suggestions:
In the introduction, we would like to thank all reviewers for reviewing our manuscript and for their valuable comments and suggestions. We revised the manuscript according to the instructions. Changes are marked in green.
Reviewer 3.
This review manuscript by Anna Wasilewska and colleagues described the function of KIR and HLA-C in infertility. The topic is of great interest and importance. The authors have reviewed a suite amount of diverse literature more or less related to the topic. Some sections of the manuscript are well-written.
Thank you for your positive opinion.
However, some may need to improve, such as:
- Section 1 may need to highlight the reasons for infertility and specific reasons for immune problems.
The text has been supplemented with immunological causes of infertility.
- Specifically, during pregnancy, most NK cells are called uNK; section 2 needs to describe the differences and functions of uNK cells.
Information regarding uNK has been added in section 3, along with a figure summarizing the subpopulations of these cells.
- Section 5 is better combined with sections 6, 7, 8, and 9 to focus on the function of KIR and HLA-C in infertility.
In our opinion, separating sections 5 and others makes it easier to find information of interest to readers, and that is why this arrangement has been left. However, if combining these sections is necessary for the publication of the manuscript, we consent to such a combination.
Thank you for your valuable time reviewing our manuscript.
Round 2
Reviewer 3 Report
Comments and Suggestions for Authors
no comment this time.